# Integrated systems analysis reveals conserved gene networks underlying response to spinal cord injury

**Jordan W Squair[1†], Seth Tigchelaar[1], Kyung-Mee Moon[2], Jie Liu[1], Wolfram Tetzlaff[1], Brian K Kwon[1,3], Andrei V Krassioukov[1,4,5], Christopher R West[1,6], Leonard J Foster[2,7], Michael A Skinnider[2†*]**

[1]International Collaboration on Repair Discoveries, University of British Columbia, Vancouver, Canada; [2]Centre for High-Throughput Biology, University of British Columbia, Vancouver, Canada; [3]Department of Orthopaedics, University of British Columbia, Vancouver, Canada; [4]GF Strong Rehabilitation Centre, Vancouver Health Authority, Vancouver, Canada; [5]Department of Medicine, Division of Physical Medicine and Rehabilitation, University of British Columbia, Vancouver, Canada; [6]School of Kinesiology, University of British Columbia, Vancouver, Canada; [7]Department of Biochemistry and Molecular Biology and Michael Smith Laboratories, University of British Columbia, Vancouver, Canada

**Abstract** Spinal cord injury (SCI) is a devastating neurological condition for which there are currently no effective treatment options to restore function. A major obstacle to the development of new therapies is our fragmentary understanding of the coordinated pathophysiological processes triggered by damage to the human spinal cord. Here, we describe a systems biology approach to integrate decades of small-scale experiments with unbiased, genome-wide gene expression from the human spinal cord, revealing a gene regulatory network signature of the pathophysiological response to SCI. Our integrative analyses converge on an evolutionarily conserved gene subnetwork enriched for genes associated with the response to SCI by small-scale experiments, and whose expression is upregulated in a severity-dependent manner following injury and downregulated in functional recovery. We validate the severity-dependent upregulation of this subnetwork in rodents in primary transcriptomic and proteomic studies. Our analysis provides systems-level view of the coordinated molecular processes activated in response to SCI.
DOI: https://doi.org/10.7554/eLife.39188.001

**\*For correspondence:**
michael.skinnider@msl.ubc.ca

†These authors contributed equally to this work

**Competing interests:** The authors declare that no competing interests exist.

## Introduction

Spinal cord injury (SCI) results in impairment of motor, sensory, and autonomic systems, causing profound deregulation of almost every bodily function. The failure of large-scale clinical trials of drug therapies in acute SCI (*Bracken et al., 1990*; *Geisler et al., 2001*), and the lack of success in translating preclinical therapies to humans (*Ramer et al., 2014*), leaves clinicians without effective treatment options for SCI. As such, hemodynamic management and surgical decompression remain the only options to influence neurological outcomes immediately following acute SCI, typically with only marginal improvements (*Hawryluk et al., 2015*; *Tee et al., 2017*; *Fehlings et al., 2012*). The absence of an effective treatment for SCI reflects the complexity of the pathophysiologic mechanisms activated by central nervous system (CNS) injury. The additive effects of the immune response (*Kigerl et al., 2009*; *Demjen et al., 2004*), multiple forms of cell death (*Springer et al., 1999*; *Crowe et al., 1997*), neuronal growth suppression (*GrandPré et al., 2000*; *Schnell and Schwab, 1990*), and the

formation of an inhibitory glial scar (*Bradbury et al., 2002*) pose a challenge to the development of new therapeutic strategies.

A major obstacle to the development of targeted therapies for SCI is the fragmentary state of our understanding of SCI pathophysiology. The response to trauma within the human spinal cord is mediated by multiple coordinated molecular pathways, yet these processes are rarely studied in an integrated manner. An additional challenge in translation of novel therapies is the reliance of clinical trials on standardized neurological assessments for patient enrolment and stratification (*Fawcett et al., 2007*). These measures are highly variable, operator-dependent, and may be impossible to perform in many SCI patients (*Kwon et al., 2017*). Systems biology approaches provide powerful means to elucidate the coordinated molecular processes underlying the pathophysiology of complex diseases (*Voineagu et al., 2011*; *Parikshak et al., 2013*; *Zhang et al., 2013*; *Johnson et al., 2016*). In particular, gene coexpression network analysis can complement reductionist descriptions of isolated gene functions by identifying networks of genes responsible for driving disease processes (*Zhang and Horvath, 2005*; *Parikshak et al., 2015*). Systems-level analyses may additionally have the potential to suggest novel biomarkers capable of stratifying injury severity and predicting functional recovery, and consequently to facilitate the translation of new therapies for acute SCI.

In the present study, we describe an integrated systems biology approach to study the pathophysiology of SCI. We systematically survey decades of biomedical literature in order to establish the complete set of genes implicated in the response to SCI by small-scale experiments. We then integrate this literature-curated gene set with unbiased gene expression data from the human spinal cord. We use weighted gene coexpression network analysis (WGCNA) to establish the normal biological processes within the healthy human spinal cord, and conduct a meta-analysis of publicly available gene expression data to define the gene regulatory network signature of the coordinated physiological response to SCI. We validate our findings at the transcriptomic and proteomic levels, and leverage the resulting systems-level understanding of SCI pathophysiology to define candidate biomarkers for stratification of injury severity and prediction of functional recovery.

## Results

### Systematic literature analysis identifies genes associated with response to SCI

Despite decades of study, an integrated understanding of the pathophysiological response to SCI remains elusive. This gap represents a central challenge to the development of targeted therapies for SCI. We hypothesized that such an integrated understanding could be achieved by integrating the vast corpus of SCI literature, collected by small-scale experimentation over several decades, within an unbiased, genome-wide framework. An overview of our experimental design is shown in *Figure 1*.

As a first step, we sought to systematically establish the complete set of genes implicated in the physiological response to SCI. We conducted a systematic analysis of the SCI literature, reviewing over 500 papers, in order to reveal a set of 695 unique human genes associated with the response to SCI by small-scale experiments (*Supplementary file 1*). Of these genes, 559 were upregulated following SCI, 213 were downregulated, and the protein products of 8 were differentially phosphorylated. Among all genes, 151 were associated with the response to SCI by more than one study (*Figure 2A*). The complete set includes genes that have been associated with SCI in a wide range of experimental models of SCI, in addition to human injuries (*Figure 2—figure supplement 1A*); in multiple species, including human as well as rat, mouse, and rabbit (*Figure 2—figure supplement 1B*); using a range of experimental techniques (*Figure 2B*); and at a variety of time points, from 1 hr to 6 months post-injury (*Figure 2—figure supplement 1C*).

### Validation of literature-curated SCI genes

We validated the biological relevance of our literature-curated (LC) SCI gene set using multiple lines of evidence. First, we established that LC genes were more likely to share common biological functions than random sets of genes, using annotations from the Gene Ontology (*Ashburner et al., 2000*). Because functional annotations may be specific or broad, we confirmed that the enrichment

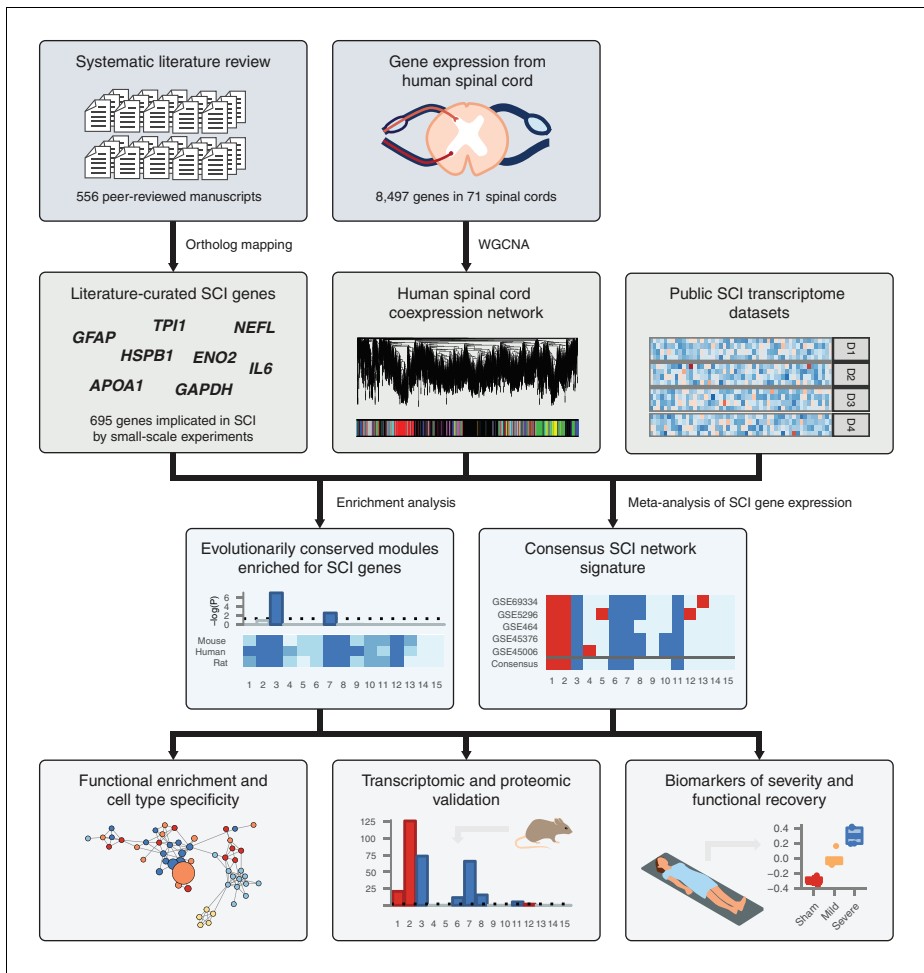

**Figure 1.** Schematic overview of systems biology approach to SCI pathophysiology integrating small-scale experiments with high-throughput data. Systematic analysis of over 500 manuscripts revealed the complete set of genes implicated in SCI pathophysiology by small-scale experiments. SCI genes were integrated with unbiased, genome-wide gene expression data from healthy human spinal cord to identify coexpressed gene subnetworks enriched for known SCI genes. Meta-analysis of SCI gene expression data revealed consensus patterns of subnetwork differential expression after SCI. The resulting consensus network signature of the response to SCI in human spinal cord was subjected to functional enrichment and cell type analyses, validated at the transcriptomic and proteomic levels, and leveraged to nominate quantitative biomarkers of SCI severity.
DOI: https://doi.org/10.7554/eLife.39188.002

held regardless of the number of genes to which each term was annotated (*Figure 2C*). Next, we investigated the tendency for the protein products of LC genes to physically interact. Significant enrichment for protein-protein interactions (PPIs) between LC genes was observed relative to random expectation (*Figure 2D*, empirical $p < 10^{-3}$), and we reproduced this finding in multiple independent PPI databases (all $p < 10^{-3}$, *Figure 2—figure supplement 2A–C*). Genes implicated in a variety of complex diseases by genome-wide association studies (GWAS) have been found to form distinct modules of densely interacting proteins within the human interactome (*Ghiassian et al., 2015*). We therefore evaluated whether this same principle held for SCI by calculating the size of the largest connected component (LCC) between LC genes, and found that LC genes collectively formed a significantly larger subnetwork than random expectation (*Figure 2E*, empirical $p < 10^{-3}$), a finding that was again reproduced in independent interaction datasets ($p < 10^{-3}$, *Figure 2—figure supplement 2D–F*). Literature-curated genes also displayed a significant tendency to participate in the same protein complexes (*Figure 2—figure supplement 2J*). Finally, LC genes were preferentially recovered by a disease gene prediction algorithm when a subset of them were randomly withheld,

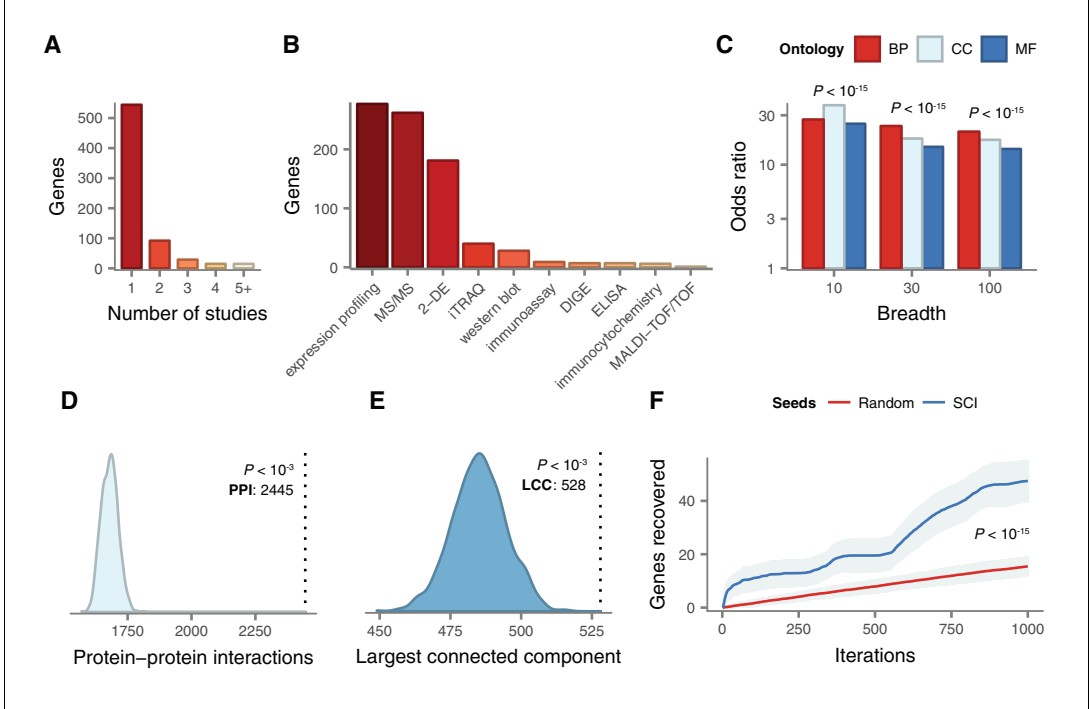

**Figure 2.** Literature curation and validation of genes implicated in the physiological response to SCI by small-scale experiments. (A) Number of small-scale studies implicating each gene in SCI pathophysiology in the LC gene set. (B) Experimental techniques used to associate LC genes with response to SCI in the LC gene set. (C) Enrichment for shared Gene Ontology terms among LC genes (all p < $10^{-15}$). BP, biological process; CC, cellular component; MF, molecular function. (D) Number of protein-protein interactions (PPIs) between LC genes observed in the high-confidence human interactome (*Menche et al., 2015*) (dotted line) and 1000 randomized interactome networks (density), revealing significant enrichment for PPIs between LC genes relative to random expectation (p < $10^{-3}$). (E) Size of the largest connected component (LCC) between LC genes in the high-confidence human interactome (dotted line) and 1000 randomized interactome networks (density), revealing LC genes occupy a distinct region of the human interactome (p < $10^{-3}$). (F) LC genes are prioritized by a disease gene prediction algorithm (*Ghiassian et al., 2015*) (p < $10^{-15}$, Kolmogorov–Smirnov test).

DOI: https://doi.org/10.7554/eLife.39188.003

The following figure supplements are available for figure 2:

**Figure supplement 1.** Literature curation of genes implicated in the physiological response to SCI.
DOI: https://doi.org/10.7554/eLife.39188.004
**Figure supplement 2.** Validation of the complete set of genes implicated in the physiological response to SCI.
DOI: https://doi.org/10.7554/eLife.39188.005

and the remainder used to prioritize additional disease genes (*Figure 2F* and *Figure 2—figure supplement 2G–I*). Thus, LC genes represent a biologically relevant and functionally coherent set of genes, which converge on a common protein interaction module within the human interactome.

## Gene coexpression network analysis of human spinal cord

Multiple lines of evidence support the functional coherence of the set of genes implicated in SCI by small-scale experiments. However, these studies nonetheless have appreciable false positive and false negative rates, and are limited by sociological and experimental biases. We therefore sought to integrate knowledge from the SCI corpus within an unbiased, genome-wide framework. We hypothesized that unsupervised gene coexpression network analysis of human spinal cord would provide a powerful method to integrate these LC genes in a systems-level context, as this method has recently been powerfully applied to develop insights into the etiologies of a number of neurological (*Langfelder et al., 2016*; *Delahaye-Duriez et al., 2016*; *Johnson et al., 2015*; *Zhang et al., 2013*) or psychiatric diseases (*Voineagu et al., 2011*; *Chen et al., 2013*; *Fromer et al., 2016*).

We constructed gene coexpression networks in human spinal cord using RNA-seq data from 71 post-mortem human spinal cords from the Genotype-Tissue Expression project (GTEx)

(*GTEx Consortium, 2013*). We applied WGCNA (*Langfelder and Horvath, 2008*) to group the human spinal cord transcriptome into 15 distinct modules of coexpressed genes (*Supplementary file 2*). These modules represent networks of genes that share highly related patterns of expression in the human spinal cord. In order to establish the reproducibility of these spinal cord gene expression modules in an independent dataset, we constructed a second human spinal cord gene coexpression network from public microarray data, using established techniques to control for batch effects (*Leek et al., 2012*; *Vandenbon et al., 2016*). Module conservation was quantified using the $Z_{summary}$ statistic (*Langfelder et al., 2011*). Despite the small sample size of our microarray-based human spinal cord coexpression network (n = 33), seven of 15 modules showed strong evidence of reproducibility ($Z_{summary} > 10$), with an additional two modules showing moderate evidence of reproducibility ($Z_{summary} > 5$) (*Figure 3A*). Only two of 15 modules showed no evidence of reproducibility ($Z_{summary} < 2$).

Next, we investigated the evolutionary conservation of human spinal cord coexpression modules in mouse and rat, two of the most commonly used model organisms for studies of SCI pathophysiology. We compiled hundreds of microarray samples from mouse (n = 414) and rat (n = 267) spinal cords from the Gene Expression Omnibus, and constructed gene coexpression networks for the mouse and rat spinal cords, again using established batch effect correction methods. Five modules showed strong evidence of evolutionary conservation ($Z_{summary} > 10$) in both species, while another four modules showed moderate evidence of conservation ($Z_{summary} > 5$) in at least one species, and

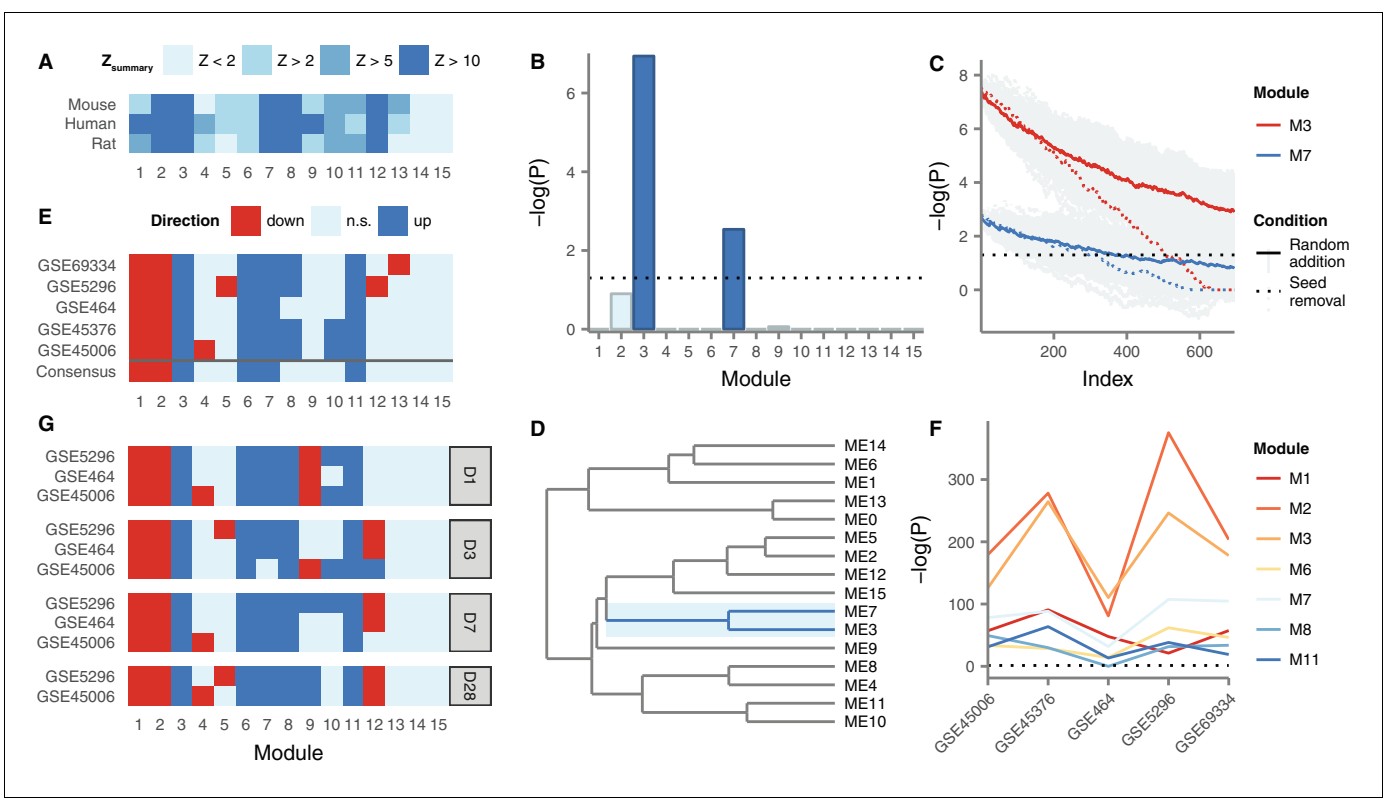

**Figure 3.** Gene coexpression modules in the human spinal cord and their differential expression in SCI. (**A**) Reproducibility of human spinal cord modules in a microarray dataset and conservation in mouse and rat. (**B**) Enrichment of M3 and M7 for LC SCI genes. (**C**) Robustness of M3 and M7 enrichment for LC SCI genes. (**D**) Eigengene network for human spinal cord modules. (**E**) Differential expression of spinal cord modules following SCI in five datasets, and consensus. (**F**) Evidence for differential expression of six consensus modules and one majority module (M8). (**G**) Time-dependent expression of spinal cord modules at acute, subacute, and chronic time points following SCI.

DOI: https://doi.org/10.7554/eLife.39188.006

The following figure supplement is available for figure 3:

**Figure supplement 1.** Contributions of experimental techniques, injury models, and species to M3 and M7 enrichment for LC genes.

DOI: https://doi.org/10.7554/eLife.39188.007

only two modules showed no evidence of conservation in either species ($Z_{\mathrm{summary}}$ < 2) (**Figure 3A**). Notably, the same five modules that showed the strongest evidence of reproducibility (M2, M3, M7, M8, and M12) also showed the strongest evidence of conservation in rat and mouse. Thus, at least at the systems level, the architecture of the spinal cord transcriptome is substantially conserved between human and model organisms, supporting our approach of integrating data from small-scale studies of mammalian model organisms.

In order to integrate the LC gene set with the spinal cord coexpression network, we next tested for enrichment of LC genes within each module (**Figure 3B**). Two modules, M3 and M7, were significantly enriched for LC genes (Fisher's exact test, Bonferroni-corrected p = $9.5 \times 10^{-8}$ and $2.7 \times 10^{-3}$, respectively). These modules consist of 746 and 330 genes, respectively, and both are among the most reproducible and conserved in the spinal cord (**Figure 3A**). We confirmed the robustness of the observed enrichment by randomly removing seed genes from the LC set, and by randomly adding false positive genes to the LC set. Both M3 and M7 remained significantly enriched for LC genes despite the removal of a large number of seed genes, or the addition of a large number of random genes (**Figure 3C**): M3 remained significantly enriched for LC genes even after the removal of approximately 70% of genes from the seed set, compared to approximately 50% for M7. Moreover, M3 remained significantly enriched for seed genes even after the size of the literature-curated set was doubled by addition of random false positives. We also asked whether the observed enrichment was driven most strongly by any individual analytical technique or injury model, but found the majority of experimental methods, SCI models, and species contributed to the observed LC gene enrichment in M3 and M7 (**Figure 3—figure supplement 1**). Thus, M3 and M7 are robustly enriched for genes associated to the SCI response by small-scale studies, despite their divergent experimental designs.

Finally, to assess the relationships between modules, we constructed a module meta-network based on the eigengene of each module, defined as the first principal component of module expression (**Langfelder and Horvath, 2007**) (**Figure 3D**). In the resulting network, M3 and M7 clustered together, as would be expected given the strong correlation between their eigengenes (Spearman's $\rho$ = 0.54, p = $1.6 \times 10^{-6}$). These results suggest that the expression of these two modules in the spinal cord is highly correlated.

In summary, gene coexpression network analysis identified five highly conserved and reproducible modules, two of which are significantly and robustly enriched for LC genes, and whose expression is highly correlated.

## Meta-analysis of coexpression network deregulation in SCI

We next characterized the role of M3 and M7, as well as other highly conserved coexpression modules, in the pathophysiological response to SCI. We performed a meta-analysis of five mouse and rat transcriptomic studies of SCI within the context of our spinal cord coexpression network, in order to identify consensus changes in the spinal cord transcriptome at the module level in response to SCI (**Figure 3E**). This analysis identified M3, M6, M7, and M11 as consensus upregulated, and M1 and M2 as consensus downregulated, following SCI. One other module, M8, was upregulated following SCI in four of five datasets, while the remaining eight modules did not show robust evidence of differential expression. Among all seven modules, M2, M3, and M7 consistently showed the strongest evidence of differential expression (**Figure 3F**, p $\leq$ $6.5 \times 10^{-36}$, $1.2 \times 10^{-48}$, and $1.6 \times 10^{-14}$, respectively). Notably, among these modules, M2, M3, M7 were strongly conserved and reproducible in mouse, rat, and human networks ($Z_{\mathrm{summary}}$ > 10), whereas M1, M6, and M11 displayed only moderate evidence of conservation (2 < $Z_{\mathrm{summary}}$ < 10), suggesting these modules may capture human-specific aspects of spinal cord transcriptome organization that are relevant in the response to SCI.

Because the pathophysiological processes underlying primary and secondary injury in SCI are incompletely understood, we additionally investigated the expression of spinal cord modules at acute, subacute, and chronic time points. Consensus module expression was remarkably consistent at all time points studied (**Figure 3G**). However, analysis of the temporal regulation of spinal cord modules revealed consensus downregulation of M9 at the most acute time point after SCI, but consensus upregulation at a chronic time point. These results suggest M9 may be specifically involved in the transition between acute and chronic physiological responses following SCI. Thus, by integrating

gene coexpression network analysis with a meta-analysis of the SCI transcriptome, we reveal a consensus network signature associated with the response to SCI, and a network module specifically implicated in the transition from acute to chronic injury processes.

## Functional characterization of consensus signature modules

We sought to characterize the biological significance of the modules implicated in the physiological response to SCI by integrating functional annotations from the Gene Ontology (*Ashburner et al., 2000*) and molecular signatures from MSigDB (*Liberzon et al., 2011*) (*Supplementary file 3*). To visualize statistically overrepresented gene sets, we constructed enrichment maps for each consensus signature module (*Merico et al., 2010*) (*Figure 4A–B* and *Figure 4—figure supplements 1– 4*). To appreciate the cell type-specificity of each module, we additionally conducted a meta-analysis of transcriptomic and proteomic profiles from the major cell types of the CNS, incorporating both bulk and single-cell RNA-seq datasets (*Zhang et al., 2014*; *Sharma et al., 2015*; *Cahoy et al., 2008*; *Zeisel et al., 2018*) (*Figure 4C* and *Figure 4—figure supplement 5*). M1 was an oligodendrocyte module, associated with axon ensheathment and myelination, whereas M2 was a neuronal module implicated in synaptic transmission. M3 was enriched for markers of microglia and vascular endothelial cells, and biological processes such as inflammatory response and response to wounding, while M7 was a microglial module enriched for annotations related to the immune response. M9 was enriched for astrocyte markers and terms such as oxidation-reduction process, as well as the term central nervous system development, which may be related to its upregulation at chronic time points following SCI. M6 and M11 were not significantly associated with any specific cell type, and were enriched for terms including cellular protein modification process and mitochondrial translation, respectively.

## Network analysis of SCI severity and recovery

The finding that M3 is a highly conserved and reproducible gene coexpression module, with the most significant enrichment for LC genes and strong evidence of upregulation following SCI, suggested that this module plays a key pathophysiological role in SCI. We focused on the role of M3 in SCI by investigating the relationship between M3 expression and two key clinical parameters in SCI: injury severity and recovery of sensory and motor function.

We first re-analysed gene expression data from a mouse model of severity-dependent injury to identify relationships between consensus module expression and injury severity (*Di Giovanni et al., 2003*; *De Biase et al., 2005*). Strikingly, M3 was the sole module enriched for genes positively correlated to injury severity, whereas M1, M2, and M9 were enriched for genes anti-correlated to injury severity (*Figure 5A*). We investigated this effect further by considering the correlations between module eigengenes, which provide a summary of the expression profile of each module, and injury severity. This analysis revealed that the M3 eigengene was the most strongly correlated with injury severity (Spearman's $\rho = 0.79$, p = $2.5 \times 10^{-7}$), with a clear separation in M3 expression between the mild, severe, and sham injury groups at 7 days post-injury (*Figure 5E*).

In order to validate the severity-dependent upregulation of M3 following SCI, we conducted a prospective experimental SCI study, using the field standard contusion injury model at the T10 segment, and performed RNA sequencing of the spinal cord parenchyma in rats subjected to moderate, severe, or sham injuries (*n* = 5 per group). Our RNA-seq data reproduced the consensus network signature derived from our meta-analysis of microarray datasets, emphasizing the robustness of this systems-level characterization of SCI pathophysiology (*Figure 5F*). In addition, we confirmed the significant association between injury severity and the M3 eigengene (*Figure 5E*; Spearman's $\rho = 0.94$, p = $4.2 \times 10^{-7}$). Thus, insights into the network-level organization of the transcriptome in SCI derived from a meta-analysis of publicly available data replicate in an independently collected dataset.

Together, these results emphasized the severity-dependent upregulation of M3 following SCI, and suggested that the expression of a gene or combination of genes that accurately summarize the transcriptional status of M3 has the potential to serve as an objective biomarker of SCI severity. To evaluate the potential of such an indicator as a biomarker of injury severity, we focused on the hub genes of M3. These genes are the most central and interconnected within the module, based on their correlation to the module eigengene, and are highly enriched for functionally relevant genes

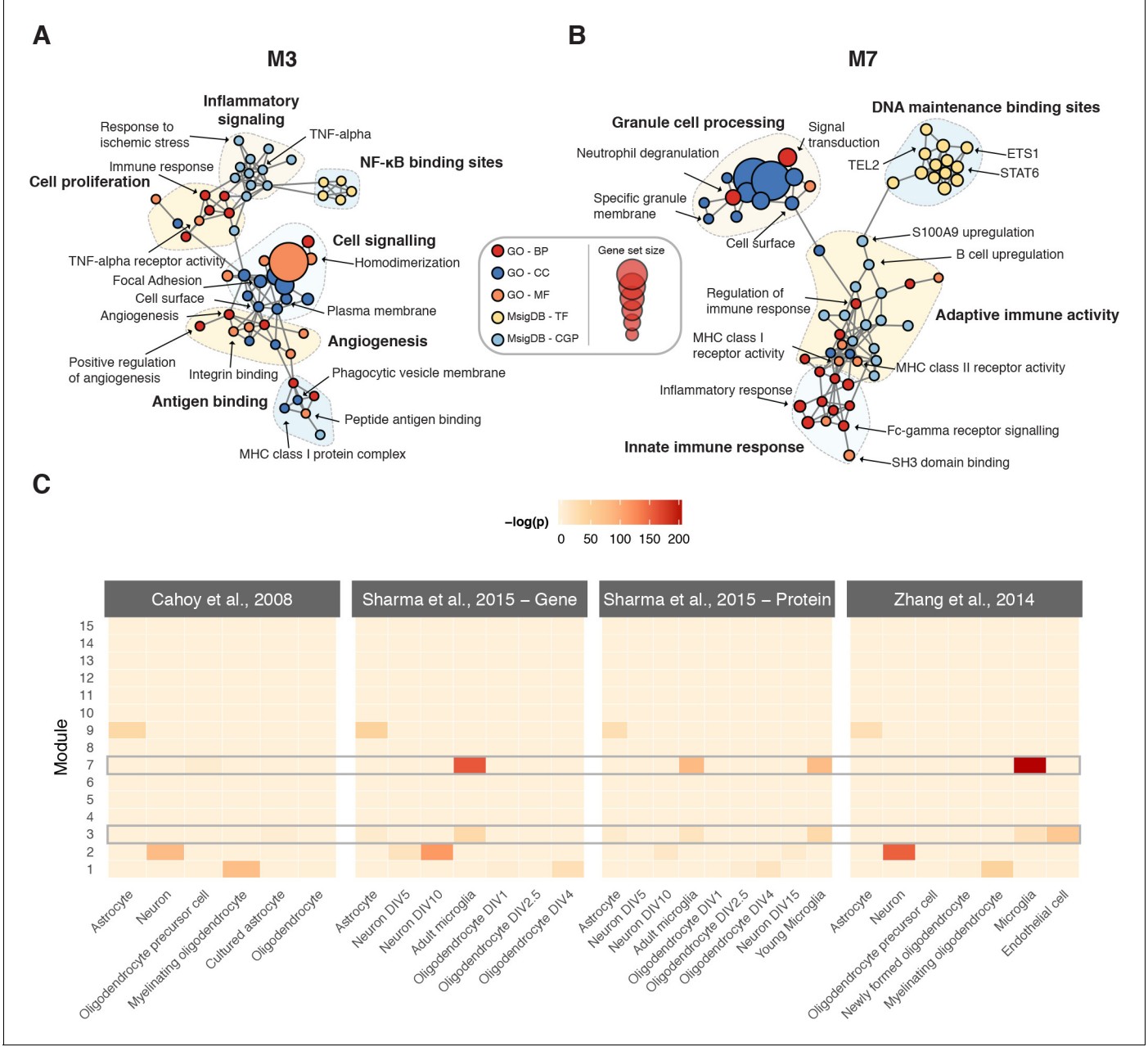

**Figure 4.** Biological characterization of spinal cord modules. (A–B) Enrichment maps (*Merico et al., 2010*) for modules M3 and M7. (C) Meta-analysis of cell type-specific marker gene enrichment in human spinal cord modules at the transcriptomic and proteomic levels.

DOI: https://doi.org/10.7554/eLife.39188.008

The following figure supplements are available for figure 4:

**Figure supplement 1.** Enrichment map for human spinal cord module M1.
DOI: https://doi.org/10.7554/eLife.39188.009

**Figure supplement 2.** Enrichment map for human spinal cord module M2.
DOI: https://doi.org/10.7554/eLife.39188.010

**Figure supplement 3.** Enrichment map for human spinal cord module M6.
DOI: https://doi.org/10.7554/eLife.39188.011

**Figure supplement 4.** Enrichment map for human spinal cord module M11.
DOI: https://doi.org/10.7554/eLife.39188.012

**Figure supplement 5.** Cell type specificity of human spinal cord modules in single-cell RNA-seq data from the mouse spinal cord (*Zeisel et al., 2018*), at three different levels of cell type classification.

*Figure 4 continued on next page*

*Figure 4 continued*

DOI: https://doi.org/10.7554/eLife.39188.013

such as drivers of disease pathophysiology (*Voineagu et al., 2011*) or therapeutic targets (*Horvath et al., 2006*). Consistent with these findings, the hubness of M3 genes (that is, their correlation to the M3 eigengene in human spinal cord) was significantly associated with their predictive power as a biomarker of injury severity (*Figure 5D*; Spearman's $\rho$ = 0.23, p = 3.9 $\times$ 10$^{-7}$). Among M3 hubs, six genes stratified rats by SCI severity with an accuracy greater than 90%, including *Anxa1*, *Colgalt1*, *Ifngr2*, *Shc1*, *Sod2*, and *Tbc1d2b* (*Figure 5G*). Remarkably, expression levels of *Anxa1* (annexin A1) stratified moderately and severely injured rats with perfect accuracy (*Figure 5I*). Annexin A1 has previously been associated with SCI by three small-scale studies, each employing divergent model organisms, spinal cord levels, and injury models, emphasizing the robustness of the association between SCI and annexin upregulation (*Didangelos et al., 2016*; *Moghieb et al., 2016*; *Gao et al., 2012*).

While our integrative analyses of public and newly acquired transcriptomic data established a strong relationship between M3 expression and SCI severity, post-transcriptional regulation can result in marked differences between gene and protein expression levels, particularly in complex tissues such as those of the CNS (*Sharma et al., 2015*; *Fortelny et al., 2017*). To further explore the potential of M3 hubs as biomarkers of SCI severity, we therefore performed quantitative proteomic profiling of the same rat spinal cords. We first sought to establish that the overall structure of the spinal cord coexpression network was conserved between the transcriptomic and proteomic levels. Despite having limited power to detect module preservation due to the small size of our proteomic sample (*n* = 15), both M3 and M7 displayed highly significant evidence of reproducibility between the RNA and protein levels (*Figure 5C*; $Z_{summary}$ = 6.8 and 7.3, respectively). Furthermore, we identified substantial overall agreement between proteomic data and the consensus network signature derived from transcriptomic meta-analysis, further validating the robustness of our systems-level portrait of SCI pathophysiology (*Figure 5F*). Finally, we confirmed the severity-dependent upregulation of both the M3 eigengene and annexin A1 in particular (*Figure 5H and J*), finding that ANXA1 protein levels stratified both moderate and severe injuries with an accuracy of 93%. Thus, systems-level insights into SCI pathophysiology derived from integrative transcriptomic analyses extend to the proteomic level and nominate quantitative biomarkers of SCI severity.

Given the strong relationship between injury severity and M3 expression, we hypothesized that targeting the transcriptional profile of this module could represent a viable strategy for development of novel therapies for SCI. To explore this hypothesis, we analyzed gene expression data from a recent trial of a neurotrophic factor, neurotrophin-3 (NT-3), which promoted sensory and motor recovery after SCI (*Duan et al., 2015*; *Yang et al., 2015*). Remarkably, all six consensus modules derived from our meta-analysis, including M3, were differentially expressed at the lesion site in the opposite direction (*Figure 5F*) in rats treated with NT-3. Intriguingly, the sole other differentially expressed module was M9, which we previously observed to exhibit a strongly time-dependent expression profile, and which was enriched for genes associated with neurogenesis. In rats treated with NT-3, known for its role in neuronal differentiation, axonal growth, and chemotropic guidance (*Alto et al., 2009*; *Anderson et al., 2016*), M9 was strongly upregulated at the lesion site relative to the experimental control (p = 9.3 $\times$ 10$^{-12}$). Moreover, the M3 eigengene was significantly downregulated in NT-3-treated rats relative to controls (*Figure 5K*; one-tailed Wilcoxon rank-sum test, p = 2.1 $\times$ 10$^{-3}$). We additionally analysed gene expression data from transgenic STAT3 knockout mice (*Anderson et al., 2016*), a loss-of-function manipulation that increased axonal dieback following experimental SCI, and found all six consensus modules were again differentially expressed in the opposite direction in wild-type mice, relative to knockout mice (*Figure 5B*). These results indicate that reversal of the transcriptome changes observed in response to SCI is associated with functional recovery and decreased axonal dieback in rodent models, and highlight M3 expression as a predictor of functional recovery.

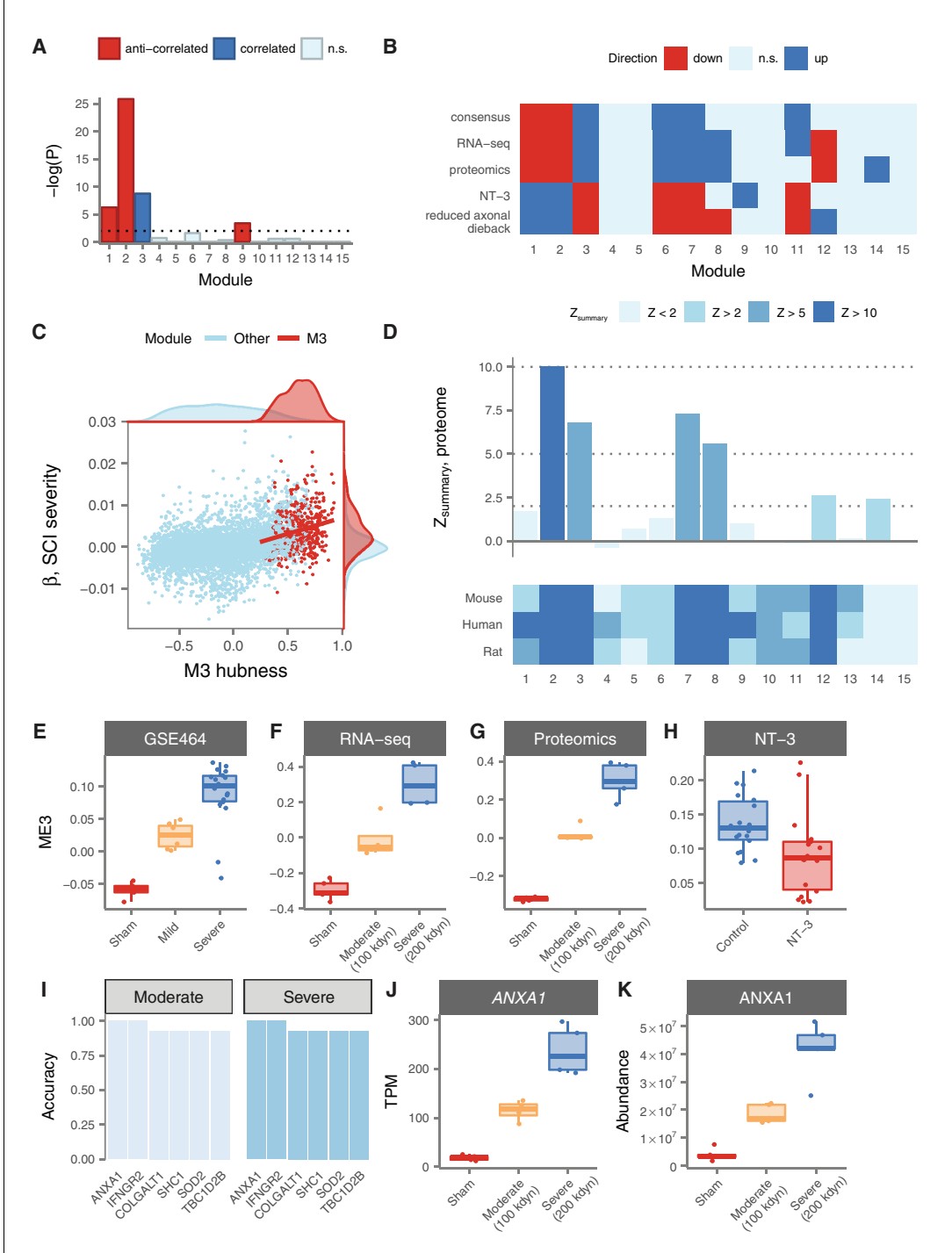

**Figure 5.** Relationship of spinal cord modules to injury severity and functional recovery. (**A**) Enrichment of spinal cord modules for genes correlated or anticorrelated to injury severity in a mouse model. (**B**) Consensus network signature of SCI pathophysiology, validation in independent transcriptomic and proteomic datasets, and reversal in functional recovery and reduced axonal dieback. (**C**) Gene expression correlation to M3 eigengene predicts association to SCI severity. (**D**) Reproducibility and evolutionary conservation of spinal cord modules and their preservation at the proteomic level. (**E–F**) Relationship between M3 eigengene and injury severity at 7 days post-injury in a mouse model (**E**), and in our own RNA-seq (**F**) and proteomic (**G**) datasets. (**H**) Downregulation of the M3 eigengene following treatment with NT-3, a neurotrophic agent that promotes functional recovery in acute SCI. (**I**) Six genes classify moderate and severe injuries in transcriptomic data with 90% or greater accuracy. (**J–K**) Gene expression and protein abundance of annexin A1 in sham, moderate, and severe SCI.

DOI: https://doi.org/10.7554/eLife.39188.014

## Discussion

The fragmentary understanding of the coordinated pathophysiological processes activated in the human spinal cord by SCI represents a central obstacle to the development of therapies capable of influencing neurological outcomes. In this study, we developed an integrated, systems-level approach to understand the molecular mechanisms underlying SCI pathophysiology. We leveraged large-scale RNA-seq data from healthy subjects to reveal gene regulatory relationships in the human spinal cord. By integrating multiple gene expression datasets from experimental models of SCI, we identified gene subnetworks implicated by consensus in the pathophysiological response to SCI, and reproduced these signatures at both the transcriptomic and proteomic levels in an animal trial. The observation that seven different gene modules were robustly associated with the response to SCI, either by consensus differential regulation (M1, M2, M3, M7, M11) or by a strongly time-dependent course of expression (M9), is consistent with the notion that the pathophysiology of SCI is highly complex (*Ramer et al., 2014*). Our results provide a framework to understand the diverse, coordinated processes in the spinal cord following SCI.

In order to prioritize gene subnetworks, we conducted a systematic analysis of the SCI literature, and integrated genes implicated in the SCI response by small-scale experiments into our network analysis. This approach is conceptually similar to the integration of GWAS or *de novo* mutation data into gene regulatory networks, as has previously been described for a number of diseases (e.g., *Johnson et al., 2015*; *Delahaye-Duriez et al., 2016*; *Calabrese et al., 2017*; *Li et al., 2014*; *International Consortium for Blood Pressure GWAS (ICBP) et al., 2015*). In the context of genetic analyses, the core assumption is that false positive and false negative associations between alleles and the phenotype of interest can be mitigated by identifying convergent molecular processes that mediate disease biology. In the context of literature curation, as employed here, we posit that the relatively high false-positive rates of small-scale experiments, as well as their appreciable false-negative rates, can be mitigated by unbiased integration of data from small-scale experiments into a genome-wide framework. Importantly, this experimental design provides an approach to extend gene coexpression network analysis to acquired and traumatic conditions, using samples from healthy tissues. However, a limitation of this approach is the implicit assumption that the molecular organization of the transcriptome in the relevant tissue of healthy human subjects is informative about the biological processes dysregulated by an acquired or traumatic condition. Although the analyses presented here indicate that this assumption appears to be valid in the case of SCI, future work will be needed to establish whether this principle holds in general.

A major challenge to the translation of preclinical therapies for acute SCI is the use of standardized neurological assessments to enrol and stratify patients in large clinical trials (*Fawcett et al., 2007*). In this context, objective biomarkers capable of accurately stratifying injury severity have the potential to facilitate translation by accelerating the pace of patient enrolment (*Kwon et al., 2017*; *Streijger et al., 2017*). We found that M3 was the sole module enriched for genes whose expression correlated with injury severity in a mouse model, and that its eigengene was likewise most strongly associated with severity. We subsequently reproduced this correlation in our own transcriptomic and proteomic datasets. The severity-dependent upregulation of M3 following SCI, and its preservation at the proteomic level, suggests that its expression has the potential to stratify injury severity in a clinical context. Furthermore, this expression pattern was reversed with administration of NT-3, a treatment that promotes motor and sensory recovery (*Yang et al., 2015*). These findings have several implications for the discovery and translation of new SCI therapies. The identification of drugs that reverse transcriptional changes associated with SCI has the potential to provide a new strategy for preclinical lead discovery. Moreover, analysing the effect of a desired treatment on M3 expression, or our consensus network signature more generally, may represent an effective technique to validate the efficacy of preclinical therapies.

Among M3 hub genes, which reflect the expression of the entire module, we found that both the RNA and protein levels of *Anxa1* (annexin A1) demonstrated a strong ability to discriminate between injuries of different severities. Annexin A1 is a member of the annexin superfamily of calcium dependent phospholipid-binding proteins, and plays a role in mediating anti-inflammatory effects through inhibition of phospholipase A2 activity (*Elderfield et al., 1993*; *Liu et al., 2007*), decreasing leukocyte activation (*Perretti and Flower, 1993*; *D'Acquisto et al., 2007*) and reducing expression of pro-inflammatory cytokines (*Sudlow et al., 1996*; *McArthur et al., 2010*). *Anxa1* is primarily

expressed in microglia, where it regulates the selective removal of apoptotic neurons (*McArthur et al., 2010*). Correspondingly, *Anxa1* knockout mice are characterized by exaggerated inflammatory responses, as well as a blunted response to the anti-inflammatory effects of glucocorticoids (*Hannon et al., 2003*). *Anxa1* is upregulated in multiple diseases characterized by aberrant neuroinflammation (*Elderfield et al., 1992*; *Elderfield et al., 1993*; *McArthur et al., 2010*). Importantly, multiple studies have previously reported upregulation of *Anxa1* in SCI (*Didangelos et al., 2016*; *Moghieb et al., 2016*; *Gao et al., 2012*), with peak expression at 7 days post injury (*Liu et al., 2004*), and upregulation of *Anxa1* is associated with functional recovery after SCI (*Liu et al., 2007*). Notably, *Anxa1* was previously identified as a biomarker of SCI severity in a study that included both rat and human samples (*Moghieb et al., 2016*). Our independent finding here that *Anxa1* is a strong candidate for a severity-dependent biomarker of SCI suggests that our systems-level approach can drive rational selection of novel potential biomarkers. However, although we observed substantial conservation of M3 between human and rat at the systems level, this finding does not preclude the possibility that individual genes diverge in their expression following acute SCI between human and rodents. Further studies in humans are therefore needed to conclusively establish the validity of *Anxa1* as a biomarker of SCI severity.

In summary, our systems biology approach identifies evolutionarily conserved and reproducible gene subnetworks with robust evidence for differential regulation following SCI, and provides a genome-wide view of the pathophysiological processes triggered by SCI. Our findings provide new, data-driven strategies to identify and translate novel therapies for SCI.

## Materials and methods

### Systematic analysis of SCI literature

We searched PubMed for articles investigating the molecular pathophysiology of SCI published prior to February 2016, using combinations of 'spinal cord injury' and one of 'proteomics,' 'proteome,' 'proteomic,' 'biomarkers,' 'biomarker,' 'RNA-seq,' and 'microarray' as search terms. 556 papers were identified that met these criteria. These were subsequently filtered to exclude papers that did not include a valid control group, included exclusively in vitro data, did not include primary data, or examined a tissue other than spinal cord. As previous studies have suggested that small-scale and high-throughput experiments may be largely complementary, or lead to divergent biological conclusions, we considered only small-scale experiments in the literature curation process, defined here as experiments reporting differential regulation of fewer than 100 genes or proteins. Ultimately, data from 67 manuscripts was collected. The original accessions used to identify genes or proteins associated with SCI in each publication were retained. If only the gene name and no unambiguous identifier was noted, the UniProt accession of the gene in the relevant species was manually retrieved. We applied a strict, majority voting-based method to map rat, mouse, and rabbit genes to their human orthologs with maximum accuracy (*Li et al., 2017*). Specifically, we mapped orthologs from rat, mouse, and rabbit genes to human using seven different ortholog databases [EggNOG (*Huerta-Cepas et al., 2016*), Ensembl (*Kinsella et al., 2011*), NCBI Gene (*Brown et al., 2015*), HomoloGene (*Agarwala et al., 2018*), InParanoid (*Sonnhammer and Östlund, 2015*), and OrthoDB (*Zdobnov et al., 2017*)], and considered human genes as 'consensus orthologs' only if they were detected in at least half of those databases containing an entry for the target model organism protein. All genes were mapped to Ensembl identifiers in Bioconductor (*Huber et al., 2015*).

### Validation of literature-curated SCI genes

We established the functional coherence and biological relevance of our LC SCI gene set using four lines of evidence: protein-protein interactions (PPIs), interactome largest connected components (LCCs), shared Gene Ontology (GO) terms, and recovery with DIAMOnD (*Ghiassian et al., 2015*), a disease gene prediction algorithm. To investigate the tendency of our LC gene set to participate in PPIs, we initially analysed a recently described high-quality human interactome (*Menche et al., 2015*). We subsequently reproduced our results in two additional human interactome databases restricted to binary and co-complex interactions, obtained from HINT (version 4) (*Das and Yu, 2012*), and a fourth human interactome, obtained from InBioMap (*Li et al., 2017*), in order to

establish that the enrichment for protein-protein interactions was not a function of the experimental technique used to detect the interaction. Self-interactions were removed and original accessions were mapped to Ensembl gene identifiers. To evaluate the impact of the experimental method used to detect PPIs, we analysed high-quality binary and co-complex interactomes from HINT separately. Network operations, including creation of induced subgraphs and calculation of largest connected components, were performed in the R package igraph (*Csardi and Nepusz, 2006*). Randomized networks were generated using a degree-preserving algorithm (*Maslov and Sneppen, 2002*) to control for network topology, with 1000 randomized networks generated to calculate empirical P values. We additionally evaluated the tendency for LC proteins to participate in the same protein complexes by retrieving random sets of proteins of equivalent size from hu.MAP (*Drew et al., 2017*) and calculating the number of co-complex interactions, a process that was repeated $10^3$ times.

GO terms were retrieved from the UniProt-GOA database (*Dimmer et al., 2012*). We compared the number of shared GO terms within each ontological category at three breadth cutoffs between all pairs of LC genes to the number of shared GO terms between random sets of genes.

To evaluate the ability of a recently described algorithm for disease gene prioritization, DIAMOnD, we randomly withheld 20% of LC genes and evaluated the fraction recovered within the first 1000 iterations of DIAMOnD. This was compared to the ability of DIAMOnD to recover an equivalent number of randomly selected genes, using randomly selected seed genes from the human genome. This process was bootstrapped 1000 times and results were reproduced using the InWeb_InBioMap interactome (version 2016_09_12) (*Li et al., 2017*)

## Gene coexpression network analysis of human spinal cord

Raw gene read count and RPKM data was downloaded from the GTEx portal, version V6p (*GTEx Consortium, 2013*). Only genes with expression estimates > 0.1 RPKM in $\geq$ 10 individuals were considered. The distribution of RPKMs in each sample was quantile transformed using the average empirical distribution observed across all samples, and inverse quantile normalization to the standard normal distribution was performed for each gene. Gene coexpression analysis was performed using the WGCNA package *Langfelder and Horvath, 2008*. Briefly, a signed pairwise gene correlation matrix was constructed using biweight midcorrelation (*Langfelder and Horvath, 2012*) and considering a maximum of 5% of samples as outliers on either side of the median. The correlation matrix was raised to the power $\beta = 5$, the minimum value satisfying the scale-free topology criterion $R^2 > 0.8$, to create an adjacency matrix. The adjacency matrix was used to calculate the topological overlap matrix (TOM), which was subsequently clustered based on the dissimilarity of gene connectivity. Coexpressed gene modules were defined from the resulting tree using the dynamic tree cut method (*Langfelder and Horvath, 2008*), with a minimum module size of 20 and a cut height of 0.2.

## Module preservation in human, rat, and mouse microarray data

We queried Array Express (*Kolesnikov et al., 2015*) with the search term 'spinal cord' to identify samples from Affymetrix GeneChip Rat Genome 230 2.0, Human Genome U133 Plus 2.0, and Mouse Genome 430 2.0 microarrays (*Supplementary file 3*). Experiments with fewer than five samples were excluded. Additionally, experiments or samples analysing individual cell populations (e.g., neurons) within the spinal cord rather than homogenized tissue, fetal spinal cord, or spinal cord neoplasms were excluded. Raw expression profiles were normalized with the MAS5 algorithm (*Hubbell et al., 2002*) within the R package affy (*Gautier et al., 2004*). Only probe sets that were called as present in at least 80% of samples were retained. ComBat was used to adjust for batch effects, where each experiment corresponded to a single batch (*Leek et al., 2012*). Affymetrix probe set identifiers were mapped to Ensembl gene accessions using Bioconductor. Genes with multiple probes were represented by the median expression value. Mouse and rat accessions were further mapped to consensus human orthologs using the majority-voting procedure described above. Module preservation was assessed using the $Z_{summary}$ statistic, calculated using the modulePreservation function in the WGCNA R package with 100 permutations.

## Meta-analysis of coexpression network deregulation following SCI

Data from five studies investigating the transcriptomic response to SCI (GSE464, GSE5296, GSE45006, GSE45376, and GSE69334) was obtained from Array Express. Normalization and mapping to human orthologs was performed as described above. For GSE464, MAS5 normalization was performed within the R package xps, as the Rat Genome U34 chip was not supported by the affy package. For GSE45376, quantile normalization of raw expression estimates was performed as described for GTEx data. Differential expression analyses of each processed dataset were performed with treatment-contrasts parameterization in the limma package (*Ritchie et al., 2015*), using the geneSetTest function to perform a mean-rank gene-set enrichment test for both up- and downregulation of each module, before applying Bonferroni correction. We additionally analysed differential expression of each module at 1 day, 3 days, 7 days, and 28 days following SCI in order to evaluate module expression at acute, subacute, and chronic time points.

## Functional characterization of spinal cord modules

To construct enrichment maps for consensus modules implicated in the response to SCI by meta-analysis, we identified overrepresented gene sets from GO (*Dimmer et al., 2012*) and the CGP (chemical and genetic perturbagens) and TFT (transcription factor targets) subsets of MSigDB (*Liberzon et al., 2011*). Genes were mapped to Ensembl accessions prior to enrichment analysis with the HTSanalyzeR package (*Wang et al., 2011*), using 1000 permutations and a minimum gene set size of 5. Significantly enriched gene sets were visualized as an enrichment map (*Merico et al., 2010*), in which nodes represent gene sets and edges connect related gene sets. Clusters of gene sets within the resulting enrichment maps were manually identified and annotated. Cell type specificity was assessed using microarray (*Cahoy et al., 2008*), RNA-seq (*Zhang et al., 2014*), and proteomic (*Sharma et al., 2015*) data from cell populations within the mouse CNS, using the R package pSI to test for enrichment of cell type-specific genes (*Dougherty et al., 2010*). We analysed a wide range of cell types, including neurons, astrocytes, and microglia (*Cahoy et al., 2008*; *Zhang et al., 2014*; *Sharma et al., 2015*), as well as oligodendrocytes at various points in maturation (i.e., 'oligodendrocyte precursor cell,' 'myelinating oligodendrocyte,' 'oligodendrocyte;' [*Cahoy et al., 2008*; *Zhang et al., 2014*]). We included additional comparisons in microglia from both new-born and adult mice (i.e., 'adult microglia,' 'young microglia;' [*Sharma et al., 2015*]), developing in vitro neurons and oligodendrocytesthrough the first two weeks of maturation (DIV: day in vitro; [*Sharma et al., 2015*]), as well as isolated and cultured astrocytes (*Cahoy et al., 2008*).

## Severity-dependent expression of spinal cord modules

Raw microarray data was obtained from Array Express (GSE464) (*Kolesnikov et al., 2015*; *De Biase et al., 2005*) and processed as described above. Injuries annotated as 'moderate' and 'severe' were grouped together on the basis of identical histological and functional outcomes in the experimental model (*De Biase et al., 2005*). To identify modules enriched for genes whose expression was correlated with injury severity, we calculated a previously described gene-level score (*Delahaye-Duriez et al., 2016*) by multiplying the Spearman correlation coefficient between gene expression and height of weight drop by the negative logarithm of the P value, then performed mean-rank gene-set enrichment tests and applied Bonferroni correction. Only samples from the lesion site were considered. Initial analysis of differentially expressed genes revealed that all but a single probe set on the U34A chip were consistently downregulated following injury, likely due to the biased composition of this chip. Therefore, samples collected using this chip were omitted when calculating correlation between injury severity and module eigengenes to reduce bias. Module eigengenes were calculated using the moduleEigengenes function from the WGCNA package.

## Spinal cord surgery and animal care

Ethical approval was obtained by the University of British Columbia Behavioural Research Ethics Board (A14-0152) and all procedures strictly adhere to the guidelines issues by the Canadian Council for Animal Care. Animals (n = 15, n = 5 per group) were started on prophylactic enrofloxin (Baytril; 10 mg kg$^{-1}$, s.c., Associated Veterinary Purchasing (AVP), Langley, Canada) three days prior to surgery. On the day of spinal cord contusion animals were anesthetized using isoflurane (initialinduction 5% and maintained on a Bain's system at 2%). After achieving surgical depth of anaesthesia, pre-

operative buprenorphine (Temgesic; 0.02 mg kg$^{-1}$, s.c., McGill University), enrofloxin (Baytril; 10 mg kg$^{-1}$, s.c., Associated Veterinary Purchasing (AVP), Langley, Canada), and Ringers solution were administered subcutaneously. The skin was prepared by shaving the surgical site, followed by three successive chlorohexidine and 70% ethanol washes. A dorsal midline incision was made from T5 to L2. The T9 spinous process was identified and a laminectomy was performed to expose the T10 spinal segment. Following this the animal was transferred to the IH impactor stage, where the T8 and T10 spinous processes were securely clamped using modified Allis forceps. The animal was stabilized on the platform and the impactor tip (2.0 mm) was properly aligned using a 3-dimensional coordinate system moving platform. The position of the impactor was confirmed as midline by two separate experimenters. The IH system was set to deliver an impact of 100 or 200 kdyn of pre-defined force, based on random assignment. Following hemostasis, the deep and superficial Para spinal muscles were sutured using 5–0 Monocryl sutures (Ethicon, USA), followed by 5–0 Prolene (Ethicon, USA) sutures in the skin. Animals were then given 5 mL of Ringers solution subcutaneously and allowed to recover in a temperature controlled environment (Animal Intensive Care Unit, Los Angeles, CA, USA). Post-operatively, animals were given buprenorphine (10 mg kg$^{-1}$, s.c.) and enrofloxin (0.02 mg kg$^{-1}$, s.c.) once daily for three days, after which buprenorphine was given on an as needed basis. Bladders were manually expressed for the duration of the experiment (7 days post-injury).

## Tissue processing

Animals were overdosed with a lethal dose of 10% chlorohydrate (i.p.), after which a thoracotomy was performed. Animals were transcardially cleared with 300 mL of PBS. Next, the spinal cord lesion site was dissected 2 mm caudal and rostral to the visual epicentre. This sample was homogenized and split into two parts for transcriptomic and proteomic processing.

## RNA isolation and sequencing

10 mg of spinal cord parenchyma surrounding the injury site was stored in RLT buffer containing beta-mercaptoethanol until RNA isolation. Total RNA was purified using the Qiagen RNeasy Mini Kit according to the manufacturer's instructions, eluting in 30 $\mu$L of water. 500 ng RNA was used for library preparation with Illumina's TruSeq Stranded mRNA kit (Illumina, San Diego, CA). Libraries were pooled and sequenced on the Illumina NextSeq in high output mode, generating paired-end 75-base pair reads. Library preparation and sequencing were performed by the Sequencing and Bioinformatics Consortium at the University of British Columbia.

## Transcriptome analysis

Quality control checks using FastQC and principal components analysis revealed a single sample as an outlier, which was discarded prior to further analysis. Sequences were pseudoaligned to the Ensembl 89 version of the Rattus norvegicus transcriptome, including coding and non-coding transcripts, with Salmon (version 0.8.2) (*Patro et al., 2017*), using 100 bootstraps to compute abundance estimates. Rat genes were mapped to human orthologs using the majority-voting procedure described above. Salmon outputwas converted into a format compatible with sleuth (*Pimentel et al., 2017*) for differential expression analysis using wasabi (https://github.com/COMBINE-lab/wasabi). Differential expression was assessed using sleuth (version 0.29.0), representing severity as a continuous covariate (0, 100, or 200 kdyn of force). Genes were ranked by the fold change estimates computed by sleuth to perform mean-rank gene-set enrichment tests for module up- and downregulation. Hub genes were identified by ranking genes by their correlation to the module eigengene in the human spinal cord samples (*Horvath and Dong, 2008*), and identifying the 10% most connected genes in the module. To evaluate the ability of hub gene expression to stratify SCI severity, we constructed linear discriminant analysis models using the R package MASS (*Kwon et al., 2017*). The accuracy of each model in classifying moderately and severely injured rats was assessed using leave-one-out cross-validation.

## Mass spectrometric analysis

Parenchyma tissue samples were lysed by 2.8 mm ceramic bead (Qiagen) prior to homogenization with a single 20 s 5000 rpm on Precellys 24 (Bertin Technologies) in 4% (w/v) SDS in 100 mM Tris pH

8.8 and 20 mM DTT. The lysate was heated at 99ᶜC for 10 min, and cell debris was spun out and protein concentration estimated by BCA assay (Thermo). An equivalent protein amount for each sample was loaded onto 10% SDS PAGE gel and visualized by colloidal Coomassie (*Candiano et al., 2004*). Each lane was fractionated into five pieces and trypsin digested out of the gel (*Shevchenko et al., 1996*). Peptide samples were purified by solid phase extraction on C-18 Stop And Go Extraction (STAGE) Tips (*Ishihama et al., 2002*), and analysed by a quadrupole–time of flight mass spectrometer (Impact II; Bruker Daltonics) coupled to an Easy nano LC 1000 HPLC (ThermoFisher Scientific) using an analytical column that was 40–50 cm long, with a 75 $\mu$m inner diameter fused silica with an integrated spray tip pulled with P-2000 laser puller (Sutter Instruments), packed with 1.9 $\mu$m diameter Reprosil-Pur C-18-AQ beads (Maisch, http://www.dr-maisch.com), and operated at 50°C with in-house built column heater. Buffer A consisted of 0.1% aqueous formic acid, and buffer B consisted of 0.1% formic acid in acetonitrile. A standard 60 min peptide separation was done per injection, and the column was washed with 100% buffer B before re-equilibration with buffer A. The Impact II was set to acquire in a data-dependent auto-MS/MS mode with inactive focus fragmenting the 20 most abundant ions (one at the time at a 18 Hz rate) after each full-range scan from m/z 200 to m/z 2000 at 5 Hz rate. The isolation window for MS/MS was 2–3 depending on the parent ion mass to charge ratio, and the collision energy ranged from 23 to 65 eV depending on ion mass and charge. Parent ions were then excluded from MS/MS for the next 0.4 min and reconsidered if their intensity increased more than five times. Singly charged ions were excluded from fragmentation.

Analysis of mass spectrometry data was performed using MaxQuant (*Cox and Mann, 2008*) version 1.5.3.30. The search was performed against a database comprised of the protein sequences from Uniprot Rattus norvegicus entries plus common contaminants with variable modifications of methionine oxidation, and N-acetylation of the proteins, and enabling LFQ and match between run options. Only those peptides exceeding the individually calculated 99% confidence limit (as opposed to the average limit for the whole experiment) were considered as accurately identified.

## Proteomic analysis

Module preservation at the proteomic level was quantified using the modulePreservation function, using Spearman correlation to calculate coexpression similarity. A mean-rank gene set enrichment test was used to test for module up- and downregulation, prior to Bonferroni correction. Linear discriminant analysis of annexin A1 was performed as described above.

## Code availability

The source code to reproduce all the analyses and generate the figures reported in this paper is available under the MIT license from https://github.com/skinnider/spinal-cord-injury-elife-2018 (*Squair and Skinnider, 2018*); copy archived at https://github.com/elifesciences-publications/spinal-cord-injury-elife-2018.

## Data availability

RNA sequencing data have been deposited to the GEO repository with the accession GSE115067. Proteomics data have been deposited to the ProteomeXchange Consortium via the PRIDE partner repository with the dataset identifier PXD010192.

## Acknowledgments

We thank Erin Erskine for assistance with animal experiments and Dr. Ward Plunet for helpful comments on the manuscript. JWS is supported by a CIHR Frederick Banting and Charles Best Canada Graduate Scholarship, a UBC Four Year Fellowship, and a Vancouver Coastal Health–CIHR–UBC MD/PhD Studentship. WT holds the John and Penny Ryan British Columbia Leadership Chair in Spinal Cord Research. BKK is the Canada Research Chair in Spinal Cord Injury and the Dvorak Chair in Spine Trauma. AVK is supported by the Canadian Foundation for Innovation, BC Knowledge Translation Foundation, the Canadian Institute for Health Research, Heart and Stroke Foundation of Canada, Rick Hansen Institute and the Craig H Neilsen Foundation. AVK holds the Chair in Rehabilitation Medicine. CRW is supported by a Scholar award from the Michael Smith Foundation for Health Research, the Heart and Stroke Foundation of Canada and the Rick Hansen Institute. LJF is supported by the Canadian Institutes of Health Research. MAS is supported by a CIHR Vanier Canada

Graduate Scholarship, an Izaak Walton Killam Memorial Pre-Doctoral Fellowship, a UBC Four Year Fellowship, and a Vancouver Coastal Health–CIHR–UBC MD/PhD Studentship. This work was enabled in part by support provided by WestGrid and Compute Canada.

## Additional information

### Funding

| Funder | Grant reference number | Author |
| --- | --- | --- |
| Canadian Institutes of Health Research | Frederick Banting and Charles Best Canada Graduate Scholarship | Jordan W Squair |
| University of British Columbia | Four Year Fellowship | Jordan W Squair<br>Michael A Skinnider |
| Vancouver Coastal Health–CIHR–UBC | MD/PhD Studentship | Jordan W Squair<br>Michael A Skinnider |
| Canadian Foundation for Innovation | | Andrei V Krassioukov |
| British Columbia Knowledge Translation Foundation | | Andrei V Krassioukov |
| Canadian Institutes of Health Research | | Andrei V Krassioukov |
| Rick Hansen Institute | | Andrei V Krassioukov |
| Craig H. Neilsen Foundation | | Andrei V Krassioukov |
| Heart and Stroke Foundation of Canada | | Andrei V Krassioukov |
| Heart and Stroke Foundation of Canada | | Christopher R West |
| Rick Hansen Institute | | Christopher R West |
| Michael Smith Foundation for Health Research | Scholar award | Christopher R West |
| Genome Canada/Genome British Columbia | 214PRO | Leonard J Foster |
| Canadian Institutes of Health Research | MOP77688 | Leonard J Foster |
| Canadian Institutes of Health Research | Vanier Canada Graduate Scholarship | Michael A Skinnider |
| Izaak Walton Killam Memorial Pre-Doctoral Fellowship | | Michael A Skinnider |

The funders had no role in study design, data collection and interpretation, or the decision to submit the work for publication.

### Author contributions

Jordan W Squair, Conceptualization, Resources, Data curation, Software, Formal analysis, Validation, Investigation, Visualization, Methodology, Writing—original draft, Project administration, Writing—review and editing; Seth Tigchelaar, Kyung-Mee Moon, Jie Liu, Data curation, Methodology, Writing—review and editing; Wolfram Tetzlaff, Brian K Kwon, Conceptualization, Resources, Supervision, Methodology, Writing—review and editing; Andrei V Krassioukov, Christopher R West, Conceptualization, Resources, Supervision, Funding acquisition, Methodology, Writing—review and editing; Leonard J Foster, Conceptualization, Resources, Supervision, Methodology, Project administration, Writing—review and editing; Michael A Skinnider, Conceptualization, Resources, Data curation, Software, Formal analysis, Supervision, Validation, Investigation, Visualization, Methodology, Writing—original draft, Project administration, Writing—review and editing

**Author ORCIDs**
Wolfram Tetzlaff [iD] http://orcid.org/0000-0003-3462-1676
Leonard J Foster [iD] http://orcid.org/0000-0001-8551-4817
Michael A Skinnider [iD] https://orcid.org/0000-0002-2168-1621

**Ethics**
Animal experimentation: Ethical approval was obtained by the University of British Columbia Behavioural Research Ethics Board (A14-0152) and all procedures strictly adhere to the guidelines issues by the Canadian Council for Animal Care.

**Decision letter and Author response**
Decision letter https://doi.org/10.7554/eLife.39188.035
Author response https://doi.org/10.7554/eLife.39188.036

# Additional files

**Supplementary files**
• Supplementary file 1. Genes associated with response to SCI in small-scale experiments, and their human orthologs.
DOI: https://doi.org/10.7554/eLife.39188.015

• Supplementary file 2. Assignment of human genes to coexpressed modules in the spinal cord.
DOI: https://doi.org/10.7554/eLife.39188.016

• Supplementary file 3. Gene Ontology terms enriched within gene coexpression modules in the human spinal cord. BP, biological process; CC, cellular compartment; MF, molecular function.
DOI: https://doi.org/10.7554/eLife.39188.017

• Supplementary file 4. ArrayExpress experiments and samples used to construct microarray gene coexpression networks for human, rat, and mouse.
DOI: https://doi.org/10.7554/eLife.39188.018

• Transparent reporting form
DOI: https://doi.org/10.7554/eLife.39188.019

**Data availability**
Sequencing data have been deposited in GEO under accession code GSE115067. They can be accessed at https://www.ncbi.nlm.nih.gov/geo/query/acc.cgi?acc=GSE115067. Proteomics data have been deposited to the ProteomeXchange Consortium via the PRIDE partner repository with the dataset identifier PXD010192. They can be accessed at https://www.ebi.ac.uk/pride/archive/projects/PXD010192.

The following datasets were generated:

| Author(s) | Year | Dataset title | Dataset URL | Database, license, and accessibility information |
|---|---|---|---|---|
| Squair JW, Tigchelaar S, Moon K, Liu J, Tetzlaff W, Kwon BK, Krassioukov AV, West CR, Foster LJ, Skinnider MA | 2018 | Integrated systems analysis reveals conserved gene networks underlying response to spinal cord injury | https://www.ncbi.nlm.nih.gov/geo/query/acc.cgi?acc=GSE115067 | Publicly available at EBI PRIDE (accession no: GSE115067) |
| Squair JW, Tigchelaar S, Moon K, Liu J, Tetzlaff W, Kwon BK, Krassioukov AV, West CR, Foster LJ, Skinnider MA | 2018 | Integrated systems analysis reveals conserved gene networks underlying response to spinal cord injury | https://www.ebi.ac.uk/pride/archive/projects/PXD010192 | Publicly available at EBI PRIDE (accession no: PXD010192) |

The following previously published datasets were used:

| Author(s) | Year | Dataset title | Dataset URL | Database, license, and accessibility information |
|-----------|------|---------------|-------------|--------------------------------------------------|
| Di Giovanni S, Knoblach SM, Brandoli C, Aden SA, Hoffman EP, Faden AI | 2003 | CNS Regeneration | https://www.ncbi.nlm.nih.gov/geo/query/acc.cgi?acc=GSE464 | Publicly available at the NCBI Gene Expression Omnibus (accession no: GSE464) |
| Faden A | 2006 | Spinal Cord Injury Murine Model | https://www.ncbi.nlm.nih.gov/geo/query/acc.cgi?acc=GSE5296 | Publicly available at the NCBI Gene Expression Omnibus (accession no: GSE5296) |
| Chamankhah M, Eftekharpour E, Karimi-Abdolrezaee S, Boutros PC, Fehlings MG | 2013 | Genome-Wide Gene Expression Profiling of Spinal Cord Injury in Rat | https://www.ncbi.nlm.nih.gov/geo/query/acc.cgi?acc=GSE45006 | Publicly available at the NCBI Gene Expression Omnibus (accession no: GSE45006) |
| Chen K, Deng S, Lu H, Wu JQ, Cao Q | 2013 | RNA-Seq characterization of spinal cord injury transcriptome in acute/subacute phases: a resource for understanding the pathology at the systems level | https://www.ncbi.nlm.nih.gov/geo/query/acc.cgi?acc=GSE45376 | Publicly available at the NCBI Gene Expression Omnibus (accession no: GSE45376) |
| Li X, Sun YE | 2015 | Expression data from adult wistar female rat 5mm spinal cord tissue | https://www.ncbi.nlm.nih.gov/geo/query/acc.cgi?acc=GSE69334 | Publicly available at the NCBI Gene Expression Omnibus (accession no: GSE69334) |

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
