## [Decision Letter]

Thank you for submitting your article "Integrated systems analysis reveals conserved gene networks underlying response to spinal cord injury" for consideration by *eLife*. Your article has been reviewed by three peer reviewers, and the evaluation has been overseen by Ole Kiehn as the Reviewing Editor and Patricia Wittkopp as the Senior Editor. The following individuals involved in review of your submission have agreed to reveal their identity: Tune Pers (Reviewer #1); Andrea Tedeschi (Reviewer #3).

The reviewers have discussed the reviews with one another and the Reviewing Editor has drafted this decision to help you prepare a revised submission.

Summary:

The study by Squair et al. use an integrative approach to identify a set of genes predictive for spinal cord injury (SCI) severity and functional recovery. They use a combination of systematic literature reviews and WGCNA on public gene databases to develop a gene set enrichment analysis and consensus SCI network conserved in many species. They then perform functional enrichment analyses using cell-based ontologies and protein network analyses to validate the selected gene sets. They identify a microglia/vascular cell network ('M3') and an innate immune response microglia network ('M7') correlated with injury severity and functional recovery, respectively. Some of the most connected genes in the M3 gene was also found to be downregulated in a dataset where neurotrophin-3 (NT3)-coupled chitosan biomaterial was grafted into a gap of completely transected rat thoracic spinal cord. The systems approach applied to spinal cord injury is novel and the pipeline is well presented and spans over a large number of relevant experimental and computational techniques, which complement each other very well. The work is important and will most probably have an impact in SCI.

Essential revisions:

The reviewers were generally very supportive of the work as presented but had a number of comments and specific request for improving the work and its impact.

A) Additional analysis:

1) The literature curated genes share more protein-protein interactions (PPI). However, biases in the PPI database towards small-scale studies could confound the analysis. The authors should redo their analysis based on PPI data restricted to interactions from large-scale screens only (e.g. from the InWeb or, if possible, HINT database). The same bias may exist for the validation analysis focusing on the size of the largest connected component and the DIAMOnD analysis.

2) The authors should attempt to provide additional cell type specificity to the modules including M3 and M7 by using single cell RNA sequencing data from the spinal cord the following studies: a) www.biorxiv.org/content/early/2018/04/06/294918, b) Neuronal atlas of the dorsal horn defines its architecture and links sensory input to transcriptional cell types. Nature Neuroscience 2018; c) Massively Parallel Single Nucleus Transcriptional Profiling Defines Spinal Cord Neurons and Their Activity during Behavior, Cell 2018.

B) Request for deposit of material:

The authors should put together a Github repository with all of the code used to perform the extensive analyses represented here. Given that the large majority of the work product here is in silico analysis of existing public resources, it seems only right for the authors to give back to the community so that each figure can be re-constructed from source code they make publicly-available. This is critical for reproducibility.

C) Comments that may be addressed in the text:

1) In genetics, less than 4% of the results from small-scale approaches (candidate gene analyses) have not been robustly replicated (Hirschhorn, Genetics in Medicine 2002). The authors should comment on: What are the potential reasons that a literature-seed gene approach seems to work well for SCI? What were the characters of the literature small-scale study approaches and experimental techniques that yielded the most predictive genes for the M3 and M7 networks – any clear trends? Finally, were the M3 and M7 networks the top results in the RNA sequencing experiment, or were there WGCNA networks (or differentially expressed genes) stronger correlated with injury severity and functional recovery than the M3 and M7 networks, which would suggest additional relevant SCI pathways currently less well captured in literature?

2) The literature curated genes coalesce on common biological functions that depends on them being accurately annotated. Could the author comment if there is bias towards literature genes even if genes have to be annotated to at least three Gene Ontology terms?

3) The authors should mention that the assumption underlying their work is that gene expression data in control individuals is indicative of processes dysregulated in SCI. Even so this assumption seems to hold true in their case it remains to be shown whether it holds true in general.

4) As most inflammatory mediators are expressed at low levels in the uninjured spinal cord, and the subsequent increase in expression is a response to injury, they are likely to correlate with SCI severity. It may therefore be suspected to find a M3 module enriched for markers of microglia positively that correlate positively with injury severity. The authors could have consider to expand the significance of their work by including additional analysis from another publicly available dataset where a different strategy was shown to promote functional recovery after SCI. In absence of this analysis the authors should discuss the general predictivity of their analysis with respect to the different modules.

---

## [Author Response]

Essential revisions:The reviewers were generally very supportive of the work as presented but had a number of comments and specific request for improving the work and its impact.A) Additional analysis:1) The literature curated genes share more protein-protein interactions (PPI). However, biases in the PPI database towards small-scale studies could confound the analysis. The authors should redo their analysis based on PPI data restricted to interactions from large-scale screens only (e.g. from the InWeb or, if possible, HINT database). The same bias may exist for the validation analysis focusing on the size of the largest connected component and the DIAMOnD analysis.

To address this concern in the revised manuscript, we have performed all three analyses (PPI, LCC, and DIAMOnD) on all four interactomes (the high-confidence interactome collected by Menche et al., 2015, InWeb_InBioMap, and both the binary and co-complex interactomes collected in the HINT database). Importantly, all of our conclusions remain unchanged regardless of which interaction database is used to perform the analysis, suggesting our results are not biased by artefacts of any interaction database.

We have revised the manuscript accordingly by adding several panels to the corresponding figure supplement (Figure 2—figure supplement 2).

2) The authors should attempt to provide additional cell type specificity to the modules including M3 and M7 by using single cell RNA sequencing data from the spinal cord the following studies: a) www.biorxiv.org/content/early/2018/04/06/294918, b) Neuronal atlas of the dorsal horn defines its architecture and links sensory input to transcriptional cell types. Nature Neuroscience 2018; c) Massively Parallel Single Nucleus Transcriptional Profiling Defines Spinal Cord Neurons and Their Activity during Behavior, Cell 2018.

The availability of single-cell RNA-seq from the spinal cord provides an opportunity to replicate our analyses of module cell type specificity, derived from bulk RNA-seq of cultured or sorted cell populations, within in vivocell transcriptomes at the single-cell level. We analyzed the cell type specificity of our spinal cord modules within the recently published Zeisel et al. 2018, dataset at several different levels of cell type classification provided by the authors. Encouragingly, we found that at all three levels, the cell types inferred from single-cell RNA-seq matched perfectly to those inferred from the meta-analysis of bulk RNA-seq presented in the original manuscript, supporting the replicability of these results. In the revised manuscript, we include these analyses in a new supplementary figure, Figure 4—figure supplement 5.

We have revised the Results section accordingly as follows:

“To appreciate the cell type-specificity of each module, we additionally conducted a meta-analysis of transcriptomic and proteomic profiles from the major cell types of the CNS, incorporating both bulk and single-cell RNA-seq datasets (Zhang et al., 2014; Sharma et al., 2015; Cahoy et al., 2008; Zeisel et al., 2018) (Figure 4C and Figure 4—figure supplement 5).”

Unfortunately, we are unable to perform the same analysis for the Häring et al. (Nature Neuroscience2018) dataset or the Sathyamurthy et al. (Cell Reports2018) dataset. In the former case, all of the cells analyzed by the authors were neuronal cells, allowing the authors to define 30 neuronal subtypes in the dorsal horn. However, the statistical method we have used to analyze cell type-specific patterns of gene expression (pSI; Dougherty et al., 2010) tests whether genes are highly expressed in one cell type, relative to all other cell types. Because our coexpression network analysis identified a single prominent neuronal module, it is challenging to link results from bulk tissue to the neuronal subtypes at the resolution afforded by the Häring et al. study, by asking whether the neuronal module genes are more highly expressed in particular subtypes of neurons relative to other subtypes of neurons, and indeed this analysis did not produce interpretable results. In the latter case, the cell type annotations for the Sathyamurthy et al. dataset have not been made publicly available to accompany the raw expression data.

B) Request for deposit of material:The authors should put together a Github repository with all of the code used to perform the extensive analyses represented here. Given that the large majority of the work product here is in silico analysis of existing public resources, it seems only right for the authors to give back to the community so that each figure can be re-constructed from source code they make publicly-available. This is critical for reproducibility.

We are happy to make all the source code that was used to perform the analyses and produce the figures publicly available under an open-source, permissive license. We have released the source code as a GitHub repository, under the MIT license, at the following URL: https://github.com/skinnider/spinal-cord-injury-*eLife*-2018. We hope that this step will allow others to reproduce and build on our work.

During the course of collecting this source code into a single repository for public distribution, we reproduced all of our analyses presented in the paper, and identified some minor inconsistencies with some of the P-values reported in the text. None of these changes materially affect our conclusions, or approach the threshold for statistical significance. However, we make note of these changes here for completeness.

We have revised the manuscript accordingly by adding a reference to the GitHub repository in the Materials and methods section:

“Code availability

The source code to reproduce all the analyses and generate the figures reported in this paper is available under the MIT license from https://github.com/skinnider/spinal-cord-injury-*eLife*-2018.”

We have also updated the following P-values throughout the Results section:

1)In the section “Validation of literature-curated SCI genes”:

“We therefore evaluated whether this same principle held for SCI by calculating the size of the largest connected component (LCC) between LC genes, and found that LC genes collectively formed a significantly larger subnetwork than random expectation (Figure 2E, empirical P < 10^-3^), a finding that was again reproduced in independent interaction datasets (*P* < 10^-3^, Figure 2—figure supplement 2D-F).”

2) In the section “Gene coexpression network analysis of human spinal cord”:

“Two modules, M3 and M7, were significantly enriched for LC genes (Fisher’s exact test, Bonferroni-corrected *P* = 9.5 × 10^-8^ and 2.7 × 10^-3^, respectively).”

3) In the section “Meta-analysis of coexpression network dysregulation in SCI”:

“Among all seven modules, M2, M3, and M7 consistently showed the strongest evidence of differential expression (Figure 3F, *P* ≤ 6.5 × 10^-36^, 1.2 × 10^-48^, and 1.6 × 10^-14^, respectively).”

4) In the section “Network analysis of SCI severity and recovery”:

“This analysis revealed that the M3 eigengene was the most strongly correlated with injury severity (Spearman’s ρ = 0.79, *P* = 2.5 × 10^-7^), with a clear separation in M3 expression between the mild, severe, and sham injury groups at 7 days post-injury (Figure 5E). […] In rats treated with NT-3, known for its role in neuronal differentiation, axonal growth, and chemotropic guidance (Alto et al., 2009; Anderson et al., 2016), M9 was strongly upregulated at the lesion site relative to the experimental control (*P* = 9.3 × 10^-12^). Moreover, the M3 eigengene was significantly downregulated in NT-3-treated rats relative to controls (Figure 5K; one-tailed Wilcoxon rank-sum test, *P* = 2.1 × 10^-3^).”

C) Comments that may be addressed in the text:1) In genetics, less than 4% of the results from small-scale approaches (candidate gene analyses) have not been robustly replicated (Hirschhorn, Genetics in Medicine 2002). The authors should comment on: What are the potential reasons that a literature-seed gene approach seems to work well for SCI? What were the characters of the literature small-scale study approaches and experimental techniques that yielded the most predictive genes for the M3 and M7 networks – any clear trends? Finally, were the M3 and M7 networks the top results in the RNA sequencing experiment, or were there WGCNA networks (or differentially expressed genes) stronger correlated with injury severity and functional recovery than the M3 and M7 networks, which would suggest additional relevant SCI pathways currently less well captured in literature?

With respect to the first question (why does our approach of literature curation work), our interpretation is that the issues that plague small-scale studies, including investigator biases, false positives, and false negatives, can be at least partially overcome by integrating the complete corpus of accumulated small-scale literature into an unbiased, genome-wide framework. In this respect, an analogy to the application of gene coexpression analysis to genetic diseases may be useful: while the true positive rate of de novomutation studies is almost certainly higher than 4%, any experimental technique to identify de novo mutations will inherently have nonzero false positive and false negative rates. The assumption made by gene coexpression analysis is that these false positives can be mitigated because they will largely contribute noise, whereas true positives will converge on common molecular processes that can be identified in an unbiased manner from genome-scale expression profiles. We have revised the Discussion to include some consideration of these points, as follows:

“In order to prioritize gene subnetworks, we conducted a systematic analysis of the SCI literature, and integrated genes implicated in the SCI response by small-scale experiments into our network analysis. […] In the context of literature curation, as employed here, we posit that the relatively high false-positive rates of small-scale experiments, as well as their appreciable false-negative rates, can be mitigated by unbiased integration of data from small-scale experiments into a genome-wide framework.”

With respect to the second question (what characteristics of small-scale experiments yield the most predictive genes for the M3 and M7 networks), we performed an additional analysis of which (a) analytical techniques, (b) experimental injury models, and (c) model organisms contributed the greatest proportions of LC genes to the M3 and M7 modules. In the revised manuscript, we include this analysis in an additional figure supplement, Figure 3—figure supplement 1.

We have revised the Results section of the manuscript accordingly as follows:

“We confirmed the robustness of the observed enrichment by randomly removing seed genes from the LC set, and by randomly adding false positive genes to the LC set. […] Thus, M3 and M7 are robustly enriched for genes associated to the SCI response by small-scale studies, despite their divergent experimental designs.”

Finally, with respect to the third question (were the M3 and M7 networks the top results in our RNA-seq experiment), there were no modules more strongly correlated with injury severity than M3 and M7, whose eigengenes were the first and second-most strongly correlated to injury severity respectively. However, two other modules (M1 and M14) were correlated to injury severity equally as strongly as M3. M1 was a highly reproducible, but somewhat human-specific module implicated by consensus in SCI and associated with oligodendrocyte function, whereas M14 was not associated with SCI and was not reproducible in the microarray dataset. In addition, one module (M6) was correlated equally as strongly as M7; M6 was a moderately reproducible, moderately conserved ‘consensus’ SCI module involved in cellular protein modification. Overall, this suggests that our network signature of SCI (Figure 5B) captures the critical severity-dependent pathways activated by experimental SCI.

2) The literature curated genes coalesce on common biological functions that depends on them being accurately annotated. Could the author comment if there is bias towards literature genes even if genes have to be annotated to at least three Gene Ontology terms?

Like any enrichment analyses involving the Gene Ontology, our validation approach to literature-curated SCI genes makes the implicit assumption that these genes are accurately annotated within the GO. In fact, however, we found that LC genes were associated with *fewer* GO terms than the proteome average (Wilcoxon rank-sum test, P < 10^-15^). Our initial requirement that LC genes be annotated to at least three GO terms to be carried forward in the GO enrichment analysis was an attempt to control for this effect, based on literature precedent (Hein et al., 2015). However, in the revised manuscript, we have redone the analysis without this requirement. We find that, although the magnitude of the enrichment is slightly decreased, our essential findings remain unchanged: LC genes are strongly and significantly enriched for shared GO terms relative to random sets of genes, implying the LC genes converge on common biological functions.

We have revised the manuscript accordingly by replacing Figure 2C, and by removing references to this requirement in the Materials and methods, as follows:

“GO terms were retrieved from the UniProt-GOA database (Dimmer et al., 2012). We compared the number of shared GO terms within each ontological category at three breadth cutoffs between all pairs of literature-curated genes to the number of shared GO terms between random sets of genes.”

3) The authors should mention that the assumption underlying their work is that gene expression data in control individuals is indicative of processes dysregulated in SCI. Even so this assumption seems to hold true in their case it remains to be shown whether it holds true in general.

We agree that the assumption that the organization of the spinal cord transcriptome in healthy human subjects is informative about the processes dysregulated in SCI is a fundamental assumption of our approach. Given the impracticality of obtaining spinal cord parenchymal tissue from human patients, this seemed to us a reasonable assumption to make, and one that was subsequently validated by the analyses presented in the paper. However, we acknowledge that this is a potential limitation of applying gene coexpression analysis to acquired or traumatic conditions using our literature curation framework more generally. We revised the Discussion accordingly to highlight this limitation, as follows:

“Importantly, this experimental design provides an approach to extend gene coexpression network analysis to acquired and traumatic conditions, using samples from healthy tissues. […] Although the analyses presented here indicate that this assumption appears to be valid in the case of SCI, future work will be needed to establish whether this principle holds in general.”

4) As most inflammatory mediators are expressed at low levels in the uninjured spinal cord, and the subsequent increase in expression is a response to injury, they are likely to correlate with SCI severity. It may therefore be suspected to find a M3 module enriched for markers of microglia positively that correlate positively with injury severity. The authors could have consider to expand the significance of their work by including additional analysis from another publicly available dataset where a different strategy was shown to promote functional recovery after SCI. In absence of this analysis the authors should discuss the general predictivity of their analysis with respect to the different modules.

To address this comment, we analyzed an additional, recently described dataset (Anderson et al., 2016), in which the authors found that knockout of STAT3 was associated with increased axonal dieback following experimental SCI. The authors performed RNA-seq of immunoprecipitated astrocytes and non-astrocytes (flow-through) separately. We re-analyzed the flow-through fraction, comparing wild-type mice to STAT3 knockout mice, and found that the resulting signature perfectly recapitulated our consensus network signature of SCI. Indeed, the WT vs. STAT3 KO signature represents the exact opposite to the severity-dependent injury signature revealed by our own RNA-sequencing experiment. Thus, we are able to independently validate our signature in a dramatically different experimental context (i.e., inhibition of astrocytic scar formation), supporting the notion that it has a general predictivity.

We have revised the manuscript accordingly by updating Figure 5B, to include the results comparing astrocyte scar formation to prevention of scarring.

In addition, we have updated the final paragraph of the Results section to include this additional comparison, as follows:

“Given the strong relationship between injury severity and M3 expression, we hypothesized that targeting the transcriptional profile of this module could represent a viable strategy for development of novel therapies for SCI. […] These results indicate that reversal of the transcriptome changes observed in response to SCI is associated with functional recovery and decreased axonal dieback in rodent models, and highlight M3 expression as a predictor of functional recovery.”